



# Spatial distribution and trends of different precipitation variability indices based on daily data in Northern Chile between 1966 and 2015

Oliver Meseguer-Ruiz[1,2], Paulina I. Ponce-Philimon[1], Jose A. Guijarro[3], Pablo
Sarricolea[2,4]

[1]Departamento de Ciencias Históricas y Geográficas, Universidad de Tarapacá, Arica, Chile

[2]Climatology Group, University of Barcelona, Barcelona, Spain

[3]State Meteorological Agency (AEMet), Balearic Islands Office, Palma de Mallorca, Spain

[4]Departamento de Geografía, Universidad de Chile, Santiago, Chile

*Correspondence to:* Oliver Meseguer-Ruiz (omeseguer@academicos.uta.cl)

**Abstract.** Northern Chile is one of the most arid regions in the world, as includes the Atacama Desert, but in high altitudes, precipitation is recorded in a very constricted season every year. This makes that water availability is one of the main concerns for policymakers. Accumulated rainfall presents very high differences from one year to another, and this makes that climate projections have a very low degree of confidence in this area. So to this region it is more interesting to study the irregularity of precipitation itself instead of the accumulated rainfall values. According to daily data of 161 meteorological stations, 4 irregularity indices of precipitation were calculated: concentration index, entropy, persistence index and fractal dimension. These indices were determined according to observed values, and then determined their spatial distribution by interpolating following multivariate regression models that consider different geographical variables such as latitude, distance to the Amazon basin, elevation, orientation and curvature. The temporal trends of each index and for each meteorological station were also calculated, and showed different results depending on the latitude and the elevation. These changes agree with the observed modifications on the intertropical atmospheric circulation and with the changes in the precipitation diurnal cycle. These results will help to improve climate projections for these region and to inform the development of water management policies.

## 1. Introduction

Rainfall in Northern Chile is highly irregular at seasonal and annual scales (Romero et al., 2013). Being this region one of the most arid areas in the world, water availability is a key topic to be taken into consideration by policymakers in charge of regional management (Sarricolea and Romero, 2015). Rainfall behaviour in this area is affected by many factors evidenced at different timescales, from daily to multidecadal (Falvey and Garreaud, 2005; Valdés-Pineda et al., 2018). Interannual variability of precipitations is mainly explained by the ENSO, associating wet periods to the cold phase and dry periods with warm events (Garreaud and Aceituno, 2001; Valdés-Pineda et al., 2015). Considering the intraseasonal



level, precipitations in Northern Chile in summer respond to an excess of latent heat in the atmosphere and incoming radiation from the ground generating vertical air instability and convection (Sarricolea and Romero, 2015). This causes the configuration of the upper levels Bolivian High (250 hPa), activating the South American monsoon, but also areas where dry seasons are absolute, with no recorded rainfall, close to the Pacific coast (Sarricolea and Romero 2015). This makes that the dry or wet characterisation of a concrete year is determined by the rainy season, from December to April. Thus, there is a clear need for a better understanding of the nature of regional rainfall which provides the natural sources of water and because of the implications of its economic impacts (Guzmán et al., 2017; Lehmann et al., 2015).

Climate change will affect arid regions in a more intense way than others (Donat et al., 2016) and their economic activities and the natural processes may be modified or compromised. In the study area, Northern Chile (17ºS – 29ºS), general warming trends may have direct implications in the hydrological cycle (Held and Soden, 2006), affecting even more the well-known situation of drought (Sarricolea and Romero, 2015; Sarricolea et al., 2017a). In this region, national policies are focused on developing economic activities that increases water demand (lithium and copper mining activities in the Atacama Desert) but indispensable to the country. A recent study (Zappalà et al., 2018) has identified permanent changes in rainfall spatial patterns in the Amazon basin linked to a northward shift of the inter-tropical convergence zone (ITCZ) and to a widening of the rainfall band in the western Pacific Ocean during the period 1979-2016.

Because of this, robustness of and confidence on future projections of precipitation in this area appear as one of the prior concerns in order to implement adequate policies and management by local governments. Global Circulation Models are not able to differentiate natural variability from anthropogenic forcing. However, even through the projected changes in precipitation patterns are consistent among the largest scales models, but show a particular degree of uncertainty on regional to local scales. This pattern shows increases in high latitudes and wet regions and decreases in dry regions (Kirtman, et al. 2013), but there is no consistence in high and dry areas, located between the two above mentioned regions, as the study area (Fig. 1 and Fig. 2). Other work showed the fact that models agreeing in small projected changes are very sensitive to their own internal variability, and they are no longer able to agree in the sign of the change (Tebaldi et al., 2011; Power et al., 2012). Some subregions are then identified where projected changes are almost zero or even smaller, located between areas where the increase or decrease projections are consistent between models. This exposes that precipitation projections are very widespread, more than it was considered in the beginning (Power et al., 2012). Moreover, the presence of the Andes Mountain Range clearly divides the study area in an eastern slope dominated by convective activity above 4500 m a s l affecting humid air masses coming from the Amazon Basin, and a western slope, where this convective mechanisms no longer operates, but where precipitation is usually recorded in the summer (rainy) season (Houston and Hartley, 2003). The role of the orography in the spatial and temporal distribution of the accumulated amounts of precipitations has been confirmed in later works, and also the influence of sea surface temperature of the equatorial Pacific Ocean (Junquas et al., 2013; Junquas et al., 2016; Segura et al., 2016).

In this area, where temperatures have experienced clear changes since 1966 (Meseguer-Ruiz et al., 2018b), it is expected that precipitation will show changes too (Trenberth et al., 2011). The variability of precipitation in Northern Chile has already been study according to monthly amounts (Houston, 2006),



evidencing a clear seasonal pattern as it has been mentioned and with hydrological consequences in the area such as the activation of ephemeral rivers. Concerning this dimension of precipitation, some changes have been identified in the referred area (Schulz et al., 2011; Sarricolea et al., 2017b). The previously exposed reasons suggest the need to face characterization of precipitation considering not only season/annual amounts, but also analysing the spatial behaviour of daily precipitation according to different indices. Focusing on this diurnal cycle, sub-daily behaviours are also identified (Endries et al., 2018; Junquas et al., 2018), exposing differences of the distribution between day and night, and also linking it to changes in the altitude where dew point is reached (Wasko et al., 2018).

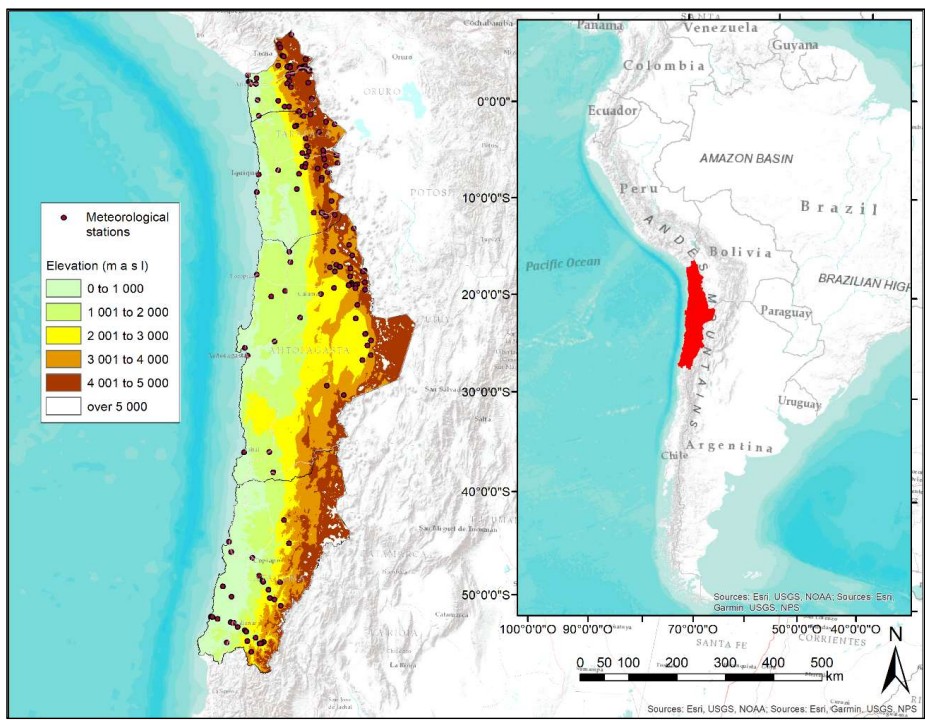

**Figure 1: Study area: orography and location of the meteorological stations.**

From a climatic point of view, it is well known that monthly amounts provide valuable information about the precipitation of a given territory. Traditionally, focus has been on means and extreme rainfall of concrete regions all around the world at global (Berghuijs et al., 2017) and regional scales: Mediterranean (Caloiero et al., 2018; Lazoglou et al., 2018; Zittis, 2017), rest of Europe (Beranová and Kyselý, 2018; Brown, 2018; Panagos et al., 2015), Africa (Abahous et al., 2017; Gummadi et al., 2017), Asia (Jiang et al., 2017; Kim et al., 2018; Ullah et al., 2018; Zhao et al., 2018), and America (Bernardino et al., 2018; Ciemer et al., 2017; Valdés-Pineda et al., 2018).

According to this, accumulated rainfall and the spatial and temporal behaviour of extreme events has been considerably well studied by the scientific community. Despite this fact, the temporal behaviour of precipitation has not been traditionally addressed as a study of irregularity, knowing that accumulated


rainfall values hide a lot of information. It is then necessary to apply other variability indices that may enlighten the temporal behaviour of precipitation and so the confidence of future climate projections may be more accurate. Defining a distribution model considering geographical variables can help to understand the spatial behaviour of an index that defines some aspects of the temporal behaviour of precipitations can

5 help to improve this.

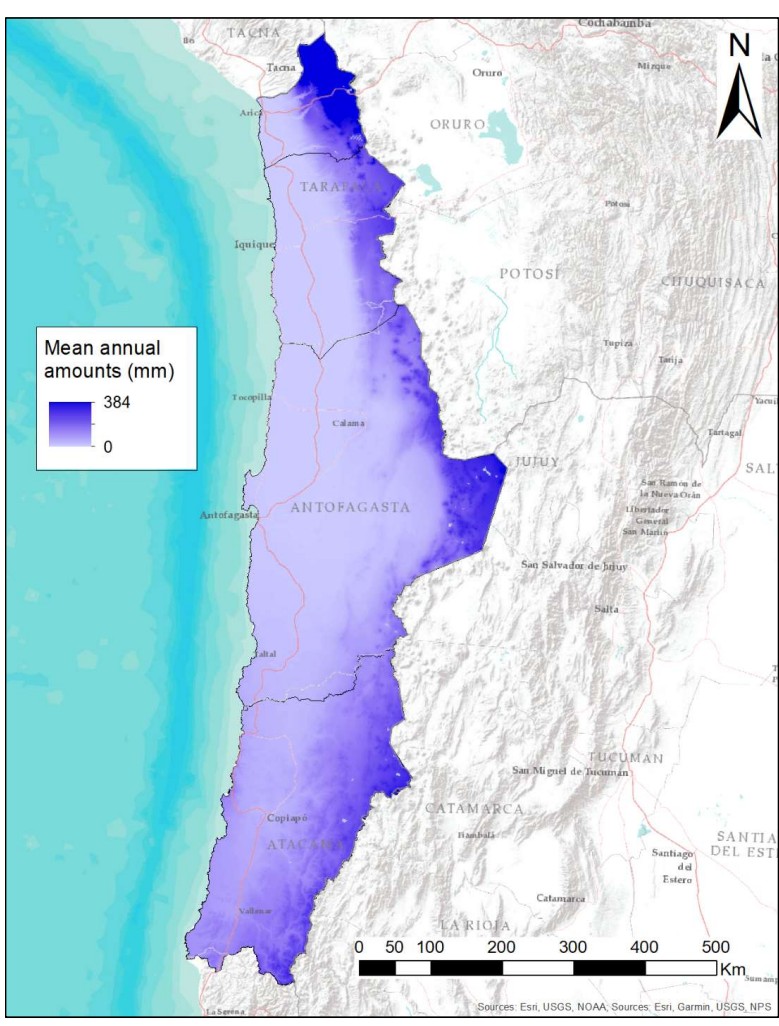

**Fig. 2. Spatial distribution of mean annual precipitation between for the 1950-2000 period (Pliscoff et al. 2014).**

Indices such as the Concentration Index (C, Martín-Vide, 2004; Monjo and Martin-Vide, 2016; Vyshkkvarkova et al., 2018) based on the Gini Index indicate the weight of the most rainy days among all the rainy days of a time series. The chronological order of the numerical values of a temporal series are not taken under consideration by many indices that quantify their variability, while it plays an essential role in



their climatic and geographical implications. The variability of a given series showed by classical statistical parameters do not varies (coefficient of variation, standard deviation…) if their records are ordered or not. The consideration of the chronological order of the records is of high interest of climate research (Meseguer-Ruiz et al., 2017). Other indices as Entropy (H) indicate the "white noise" appearing in a series, which is linked to the fractal dimension (D), showing the self-similitude characteristic of a variable (Meseguer-Ruiz et al., 2017). Finally, the Persistence Index ($P_{11}$) represents the probability of occurrence of wet spells longer than two or more days (Martín-Vide and Gomez, 1999).

The aim of this manuscript is to determine the spatial distribution of four well-known variability indices (C, H, D and $P_{11}$) in Northern Chile, an area where climate projections are not able to determine a clear trend for precipitation. The spatial distribution of these indices will be obtained according to a multivariate regression model that will consider the geographical characteristics of the area. Determining their trends across the study period (1966-2015) can help to improve this, by helping to discriminate the natural variability from possible attributions to human forcing. The manuscript states as it follows: sect. 2 explains the used dataset and methodology, sect. 3 shows the obtained results; which are discussed in sect. 4, after which appear the conclusions in sect. 5.

## 2. Material and methods

### 2.1. Observed rainfall data

Daily rainfall records from 161 stations across the study area for the period 1st January 1966 to 31st December 2015 are used.

The quality control was developed using the R package Climatol version 3.0 (Guijarro, 2016), which uses normal ratio values (every data are divided by the mean of its series) of the closest precipitation data to build reference series for all the stations. Differences between observed and reference series serve to test their quality by outlier detection, and also to check their homogeneity through the SNHT test (Alexandersson, 1986). At the same time, undoing the normalization of the reference series provides estimations to fill any missing data in the series. The detection of significant shifts in the mean was done on the monthly aggregates of the series, since the much higher variability of the daily series makes that detection far more difficult, especially in such arid climates as the one studied in this work.

A conservative approach was used by applying a threshold of SNHT=25 for the detection of shifts in order to avoid false positives, and then 32 break-points were identified with this criterion. Daily series can be adjusted at the same dates and reconstructing complete series according to the list of the break-point dates. However, due to the extreme aridity of most of the studied territory, most daily series have means lower than 1 mm, limiting the applicability of the normal ratio approach. Therefore, no adjustment was finally applied to the daily series, which were only completed by infilling all their missing values. Only the closest reference data item at every time step was used in this process to avoid the smoothing effect that would have derived from the use of several reference data, which in an arid climate with isolated precipitations would have increased the number of rainy days at the cost of averaging the values of rainy and rainless spots in the vicinity. No outstanding outliers were identified through this process that could reliably be identified as errors, and therefore all original data were kept.


### 2.2 Precipitation variability indices

#### 2.2.1. Concentration Index

The C has been calculated following the methodology shown in the original paper (Martín-Vide, 2004). It expresses the weight of the days that recorded the higher (lower) precipitation amounts. The C is an index

that varies from 0 to 1. It is related to the area between the exponent curve and diagonal of the square with a side of 100 units and 10 000 area units. The C is calculated as the proportion of the aforementioned area under the diagonal (S'/5 000). Its calculation requires the area below the exponential curve (A') (Fig. 3). This exponential adjustment relates the percentage of rainy days and their total amounts, both accumulated. The C calculation can be expressed mathematically as follows:

$$Y = aX \, exp(bX) \tag{1}$$

Where a and b are constants that can be determined through the least squares procedure following equations 2 and 3, with N representing the number of classes:

$$ln\,a = \frac{\sum X_i^2 \sum lnY_i + \sum X_i \sum X_i lnX_i - \sum X_i^2 \sum lnX_i - \sum X_i \sum X_i lnY_i}{N \sum X_i^2 - (\sum X_i)^2} \tag{2}$$

$$b = \frac{N \sum X_i^2 \sum lnY_i + \sum X_i \sum ln \;_i - N \sum X_i \, ln X_i - \sum X_i \sum lnY_i}{N \sum X_i^2 - (\sum X_i)^2} \tag{3}$$

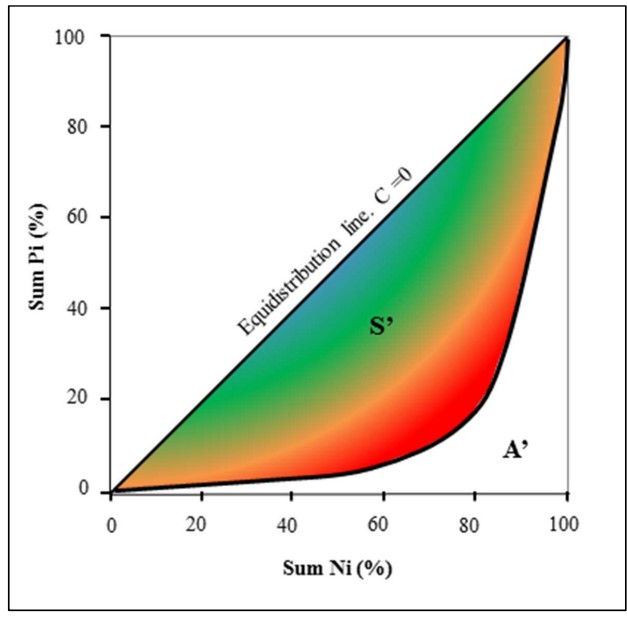

**Figure 3: Exponential curve of the cumulative number of precipitation days (Sum Ni) versus accumulated rainfall (Sum Pi)**

Once a and b (constants) are calculated, the integral of the exponential curve is defined between 0 and 100, which represents the area under the curve, A' (Equation 4 and Fig. 3):





$$A' = \left[ \frac{a}{b} e^{bx} \left( x - \frac{1}{b} \right) \right] 100 \tag{4}$$

The area over the curve and under the equidistribution line S' (Figure 2) is obtained through the 5000 area
units minus A':

$$S' = 5\,000 - A' \tag{5}$$

The C is defined as the concentration of daily precipitation in equation 6 (simplified in equation 7):

$$C = \frac{2S'}{10,000} \tag{6}$$

$$C = S'/5\,000 \tag{7}$$

### 2.2.2. Entropy

Shannon (1948) introduced the notion of entropy to determine the white noise (or disorder) of a series,
without considering its own variability. It can be applied to any climatic variable, as precipitation, as any
studies show that its distribution in recent years has become more irregular due to climate change and
intensive human activity (Liu et al., 2013). The hydrological cycle is sensitive to this, so its calculation can
help in the task on managing water resources, so its calculation and evolution can help to achieve this goal.
The entropy is calculated as it follows:

$$H(x) = - \sum p(x_i) \cdot log_2 p(x_i) \tag{8}$$

where $p_i$ is the probability of character number $i$ appearing in the stream of characters of the message.

### 2.2.3. Persistence index

The Persistence Index $P_{11}$ is defined as the probability of a rainfall episode occurring (in this case, every
day) after another rainy episode (Martín-Vide and Gomez, 1999). $P_{11}$ refers to the permanent happening
(persistence) of a rainfall event. It is calculated as it follows:

$$P_{11} = \frac{n}{p} \tag{9}$$

where $p$ is the total number of intervals where some precipitation has been recorded and $n$ is the number of
intervals where some precipitation has been recorded preceded by another precipitation interval.

### 2.2.4. Fractal dimension

The fractal dimension (D) express the regular recurrence of precipitation, i.e. that the rain episodes are
repeated regularly (or not) on different timescales (Ghanmi et al., 2013; Meseguer-Ruiz et al., 2017; 2018a).
D can be calculated following the Hurst exponent method (equation 10), because it is directly related to it
(Gneiting and Schalter, 2004):

$$D = n + 1 - H \tag{10}$$

where $H$ is the Hurst exponent and $n$ is the number of dimensions of the considered space (in this case, 1
for temporal series). In this case, D has been determined using the box-counting method. Considering the
daily records, periods containing 1, 2, 4, 6, 8, 12 and 24 days were established, and then identified in how
many of those any precipitation was recorded. Doing this, it is possible to identify the daily to sub-monthly
temporal behaviour of precipitation. The D value is determined by the slope of the regression line
representing the natural logarithms of the pairs of the length of the interval (l) and the number of intervals



with precipitation (N). D is given by 1 + α, where α is the absolute value of the slope of the regression line (Fig. 4).

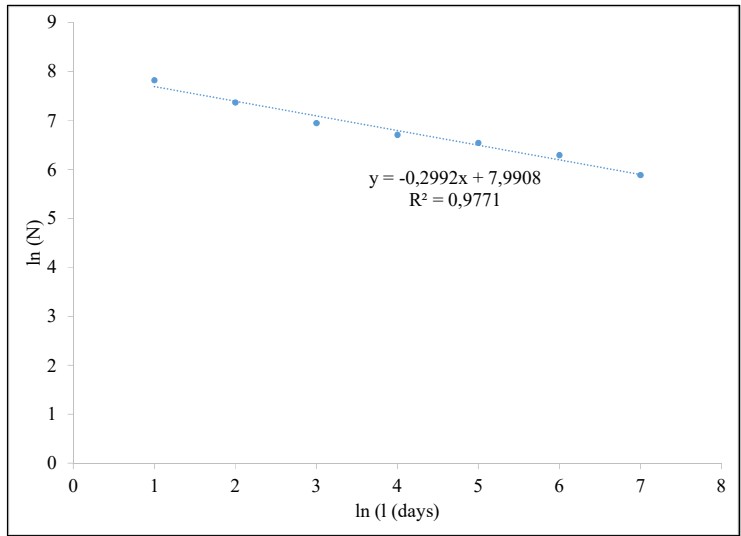

**Figure 4: Regression line for obtaining D in an hypothetical station**

**2.3. Spatial distribution of variability indices**

To estimate the spatial distribution of the different precipitation variability indices (C, F, $P_{11}$ and D) according to the spatially discrete information provided by the meteorological stations we used multivariate regression models (Brown and Comrie, 2002). This information is based on continuous variables obtained from the Shuttle Radar Topography Mission (SRTM), such as elevation, curvature and orientation (Ninyerola et al., 2000; 2007), the latitude and the distance to the Amazon basin. The spatial resolution of these continuous variables has a spatial resolution of 90 m x 90 m.

Attending that multivariate regression models generate residuals spatially distributed (differences between the meteorological stations and the values determined by the models), they were corrected following an Inverse Distance Weighting interpolation (IDW) with a power magnitude of 2 (Chen et al., 2017). This allows to obtain interpolations that approach in a better way to the spatial heterogeneity, by decreasing the mistake of the model and deleting the outliers (Crespi et al., 2018).

The multivariate regression models will be expressed according to the variables shown in Table 1, and a constant:

**2.4. Variability indices trends**

The analysis was conducted for the period 1966-2015; therefore, the annual variability indices values found were obtained over a broad period of 50 years, which gives climatic significance to the results. The trends are determined applying the Mann-Kendall non-parametric test (Mann, 1945; Kendall, 1962). The MK statistic is calculated as:





$$S = \sum_{i=1}^{n-1} \sum_{j=i+1}^{n} sgn(x_j - x_i) \tag{11}$$

$$(x_j - x_i) = z \tag{12}$$

$$sgn(z) = \begin{cases} 1 \text{ if } (z) \geq 0 \\ 0 \text{ if } (z) = 0 \\ -1 \text{ if } (z) \leq 0 \end{cases} \tag{13}$$

Where *n* is the dimension of the series and $x_j$ and $x_i$ are the annual values, respectively, in the years *j* and *i*, with *j* > *i*. For *n* > 10, given that $x_i$ is an independent and randomly ordered series, the statistic *S* follows a normal distribution whose mean is equal to 0, and variance given by:

$$Var(S) = [n(n-1)(2n+5) \sum_{i=1}^{n} t_i i(i-1)(2i+5)]/18 \tag{14}$$

Where $t_i$ represents a margin of error of *i*.

The standardized statistical test $Z_{MK}$ follows a standard normal distribution, and is represented by:

$$Z_{MK} = \begin{cases} \frac{S-1}{\sqrt{Var(S)}} \text{ if } S > 0 \\ 0 \text{ if } S = 0 \\ \frac{S+1}{\sqrt{Var(S)}} \text{ if } S < 0 \end{cases} \tag{15}$$

Using a two-tailed test, if $Z_{MK}$ is greater than $Z_{(\alpha/2)}$, with a significance level $\alpha$, then it is possible to reject the null hypothesis and the trend can be considered significant.

## 3. Results

### 3.1. Multivariate regression models

The multivariate regression models stated as shown in equations (17) to (20).

$$C = -3.4731 \cdot 10^{-6} \cdot E_{lev} - 7.919 \cdot 10^{-3} \cdot D_{AB} - 7.81 \cdot 10^{-5} \cdot O_{ri} - 0.0156 \cdot C_{urv} + 0.0142 \cdot L_{at} + 0.4921 \tag{16}$$

$$H = 2.3074 \cdot 10^{-4} \cdot E_{lev} + 0.6768 \cdot D_{AB} + 4.5735 \cdot 10^{-4} \cdot O_{ri} + 0.1680 \cdot C_{urv} - 0.7087 \cdot L_{at} + 10.6429 \tag{17}$$

$$P_{11} = -8.056 \cdot 10^{-6} \cdot E_{lev} - 0.0191 \cdot D_{AB} + 0.0184 \cdot L_{at} + 0.9386 \tag{18}$$

$$D = 2.349 \cdot 10^{-5} \cdot E_{lev} + 0.0657 \cdot D_{AB} + 2.9449 \cdot 10^{-5} \cdot O_{ri} + 0.0305 \cdot C_{urv} - 0.0837 \cdot L_{at} + 1.5846 \tag{19}$$

The regression statistics for each model are shown in Table 2.

The C shows a direct relationship with the latitude and inverse relationships with the elevation, the distance to the Amazon basin and the orientation and curvature of the surface. On the contrary, the H model presents a direct relationship with the elevation, the distance to the Amazon basin, the orientation, the curvature, and an inverse relationship with the latitude. The $P_{11}$ index is directly related with the latitude, indirectly related to the elevation and the distance to the Amazon basin and is independent from the orientation and curvature




of the surface. Finally, D is directly related to the elevation, the distance to the Amazon basin, the orientation and curvature, and indirectly to the latitude.

### 3.2. Spatial distribution of the precipitation variability indices and their trends.

The spatial distribution and trends of the C between 1955 and 2015 appear in Fig. 5. The highest C values, reaching 0.67, are found in the centre-south of the study area. To the south, these values decrease lightly, but they decrease strongly to the north, up to 0.42. In the northern half of the study area, north 23º S, the values (observed and modelled) are always below 0.59. This clear latitudinal pattern is lightly modified by the highest elevated areas, where the rainiest areas are located. The highest C values in the centre coincide with low rainy areas, between 1 000 and 3 000 m a s l, and the C are low in similar areas located in the north.

In the southern part of the area, south 23º S, no significant trends of the C values have been identified between 1966 and 2015. All the significative trends are located in the intertropical area of the study region, where positive and significative trends dominate over negative and significative trends (28 positive and 11 negative). All the significant and negative trends are located in high elevated areas, over 3 900 m a s l, meanwhile, the significant and positive trends are found with no dependence of the elevation (from 100 m a s l to 4 185 m a s l).

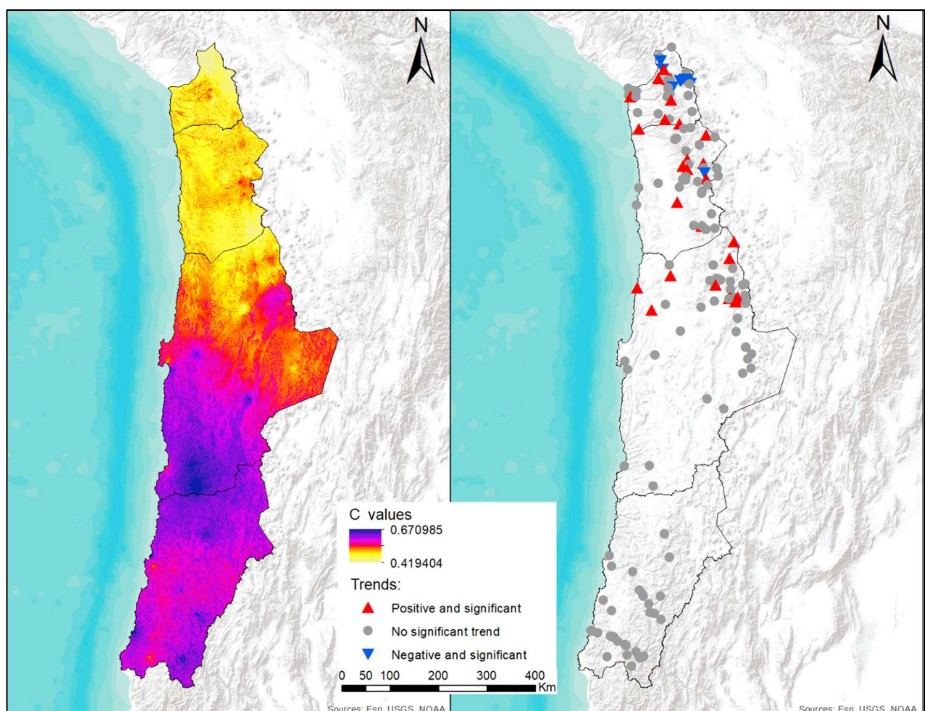

**Figure 5: C spatial distribution and trends between 1966 and 2015. The significance level considered is 90%.**





Figure 6 shows the spatial distribution of H and the observed trends between 1966 and 2015. The lowest values of H appear in the south, about 7.7 up to 8.8 in the central latitudes of the study area. Between those latitudes, there is a clear increase of H from south to north. Northern 23º S, there is a higher latitudinal

gradient to the north, where H reaches 12.0, with also a clear direct influence of the elevation (higher values in the East, in the Andes mountain range, and lower values in coastal areas). The highest values are found in the Altiplano, with elevations above 4 000 m a s l.

The H index shows 17 meteorological stations with negative and significant trends, 16 of them located at or northern the 23º S. The negative trends are found in the intertropical region of the study area, except for

one, located in the south. Otherwise, 22 stations present positive and significant trends, 14 of them are located in the south, 6 in the centre-north, and 2 in the extreme north of the study area. In the south, there is no clear difference between high or low altitudes, but in the centre and north, they are mostly located in high areas. Summarising, in the south, low H values tend to rise, but in the north, such a distinction cannot be made, while stations with high H values show increasing or decreasing significant trends.

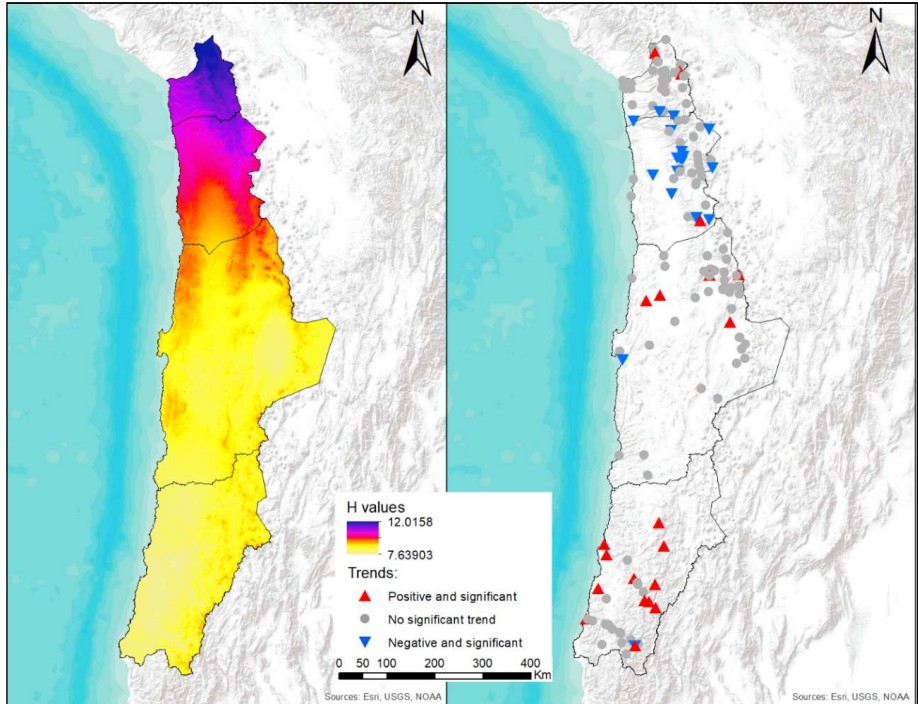

**Figure 6: H spatial distribution and trends between 1966 and 2015. The significance level considered is 90%.**

The $P_{11}$ index distribution shows a high degree of heterogeneity, no clear spatial patterns are appreciated (Fig. 7). In the north of the area, the lowest values (between 0.85 and 0.88) can be found in the eastern zone, the closest area to the Amazon basin. Just a few kilometres south, high values appear (higher than 0.95),





and the $P_{11}$ values also rise to the west, event in coastal zones (0.90 and 0.91). South from here, again low values appear, between 0.87 and 0.89, coinciding with the driest regions of the study area. From 23º S to the south, the values remain constant and elevated (from 0.95 to 0.99), and in the subregion it seems to be a very clear influence of the elevation, as in highlands, $P_{11}$ values are lower, as can be seen in the south.

5  The trend analysis show 13 stations with negative and significant trends and 25 stations with positive significant trends. In the south, 7 negative significant trends are shown, spread out from the coast to the Andes, and 1 positive trend located at 4 150 m a s l. In the northern half, in the intertropical region of the study area, 5 negative significative trends are shown. The positive and significative trends are all located in the intertropical subregion. Three of them are in the coast, and 15 appear in the Andes range and pre range, over 3 000 m a s l. But in high altitudes, 5 negative and significant trends appear too, over 4 000 m a s l.

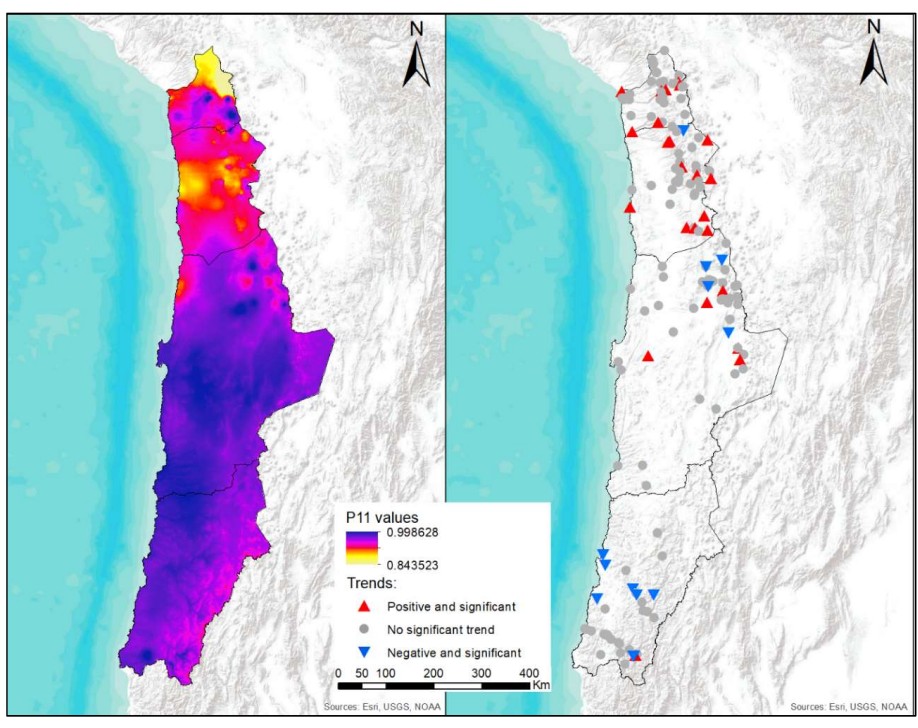

**Figure 7: $P_{11}$ spatial distribution and trends between 1966 and 2015. The significance level considered is 90%.**

The D index spatial distribution and trends is shown in Fig. 8. The highest D values, over 1.66, are found in northeast of the study area, in the Altiplano, over 4,400 m.a.s.l. These values decrease to the west, while they reach the coastline with values around 1.6, and to the southern limit of the study area, where the lowest values are found, around 1.02. This latitudinal gradient is modified by the influence of the high elevated

20  areas in the central region, but this influence disappears in the south.

Three stations show negative and significant trends, all located in the north, both in high and low altitudes, and 27 stations have a positive and significant trend. Only one of them is located in the north, the rest appear

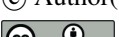



in the central region (7) and in the south (19). In this case, no differences between elevated and sea level located stations appear.

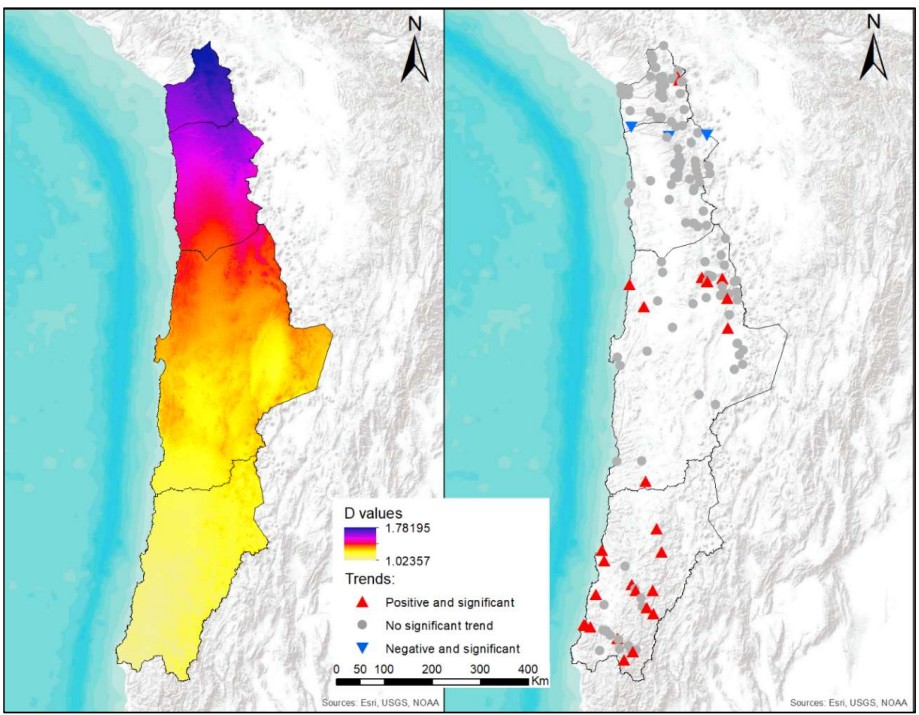

**Figure 8: D spatial distribution and trends between 1966 and 2015. The significance level considered**
**is 90%.**

## 4. Discussion

The spatial distribution of 4 irregularity of precipitation indices was determined for Northern Chile. This distribution, determined according multivariate regression models based on observed data of 161 meteorological stations, takes into consideration 5 geographical variables: elevation, distance to the
Amazon basin, latitude, orientation and curvature.

The C is negatively related to the elevation, so that means that it decreases when altitude rises to the east while the Andes mountain range appears. In these high elevated areas, the accumulated rainfall is higher, so that means that precipitation spreads into more rainy days, which agrees with Falvey and Garreaud (2005). C also increases to the south, explained by the rise of accumulated rainfall and of the number of
rainy days (Schulz et al., 2011), not only in austral summer, but also in the rest of the year (Sarricolea and Romero, 2015). The C shows different behaviours in other desert areas, according to a coastal or inland location, agreeing with the findings of Monjo and Martin-Vide (2016). In the northern region, the main source of moist air that originates the precipitation is the Amazon air mass, but to the south, other mechanisms can trigger rainy events, as cold fronts coming from the Pacific Ocean (Valdés Pineda et al.,
2015). These reasons also explain the spatial distribution of $P_{11}$. Rainfall is more persistent in the south,





while increasing latitude, where the cold fronts coming from west can also affect, but less persistent close to the Andes, while these fronts lose moisture and while they arrive to the inland, some of them may spent all of their water vapour. The highest H values are found in the northeast, where the annual accumulations are higher and very concentrated in the rainy season. These high values (over 11) means a high disorder

degree, and are consistent with the data recorded at a daily resolution, so days with rainfall records are fewer that those without and showing a high disorder. In the south, rainfall is less concentrated, the disorder is lower, explained by H values about 8 and 9. This index allows a regionalization non provided by the other indices, i.e. higher values in the north of the study area, both in the coast and inland, allow to identify a higher disorder of precipitation, corresponding in this last case with the most rainy zone of the study area.

So, the higher (lower) the number of rainy days, the lower (higher) the disorder and lower (higher) H values (Meseguer-Ruiz et al., 2017). The D index is an indicator of the regular recurrence of precipitation, i.e. that periods with precipitation are repeated regularly over time on different timescales. The spatial distribution of D are consistent with this definition, higher values in the north where a clear rainy season repeats every year between the same months (Sarricolea and Romero, 2015), and in the south, other mechanisms may

spread this phenomena during the year, making rainfall less recurrent (Meseguer-Ruiz et al., 2017; Valdés-Pineda et al., 2015).

The spatial distribution of temporal behaviour of C shows that all the stations showing significative trends are located in the intertropical area. A major number of meteorological stations show positive trends, which means that rainfall concentrations are getting higher, related with a higher convective activity (Martín-Vide,

2004), but also with a shorter rainy season. The increase of convective activity in the north agrees with the rise of the temperatures identified in this area (Meseguer-Ruiz et al., 2018b). The significative trends of the H index show negative values in the intertropical region, which would disagree with the hypothesis of the shortening of the rainy season in the north, so the lower H is, lower is the degree of disorder introduced by more rainy days (Meseguer-Ruiz et al., 2017). The same idea is reinforced by the trends showed for the

north by $P_{11}$, where the probability of a rainy day after another rises, showing a higher persistence of the rainy conditions. In the south, H decreases and $P_{11}$ increases, which means that mechanisms originating rain present a more chaotic temporal behaviour. These results disagree with other findings (Meseguer-Ruiz et al., 2017), but this can be explained because of the difference of the temporal resolution of the original data (10-minutes data against daily data). The D index rises for the whole extratropical area, which can be

explained by a more important influence of frontal precipitation against convective in this area (Ghanmi et al., 2013; Meseguer-Ruiz et al., 2018b).

C shows a spatiotemporal behaviour without any correlation with climatic zones identified in Sarricolea et al. (2017). This is related, first of all, in semiarid areas (Altiplano) with a higher concentration of precipitation in less days, as explained before. The rest of the area is dominated by the absolute desert,

where very few days of precipitation are recorded between 1966 and 2015, and an increase (or decrease) of 1-2 days of precipitation may have statistical significance in the trends. H seems to be more related to the climatic classification of the study area, but also sensitive to the modifications introduced by the latitude, as shown in the multivariate regression models. $P_{11}$ just shows consistent behaviour patterns in the south of the study regions (both spatial and temporal). In this area appears the northern border of the semiarid region

(Sarricolea et al., 2017), so precipitation is a more frequent phenomena than in the absolute desert or the




Altiplano, where precipitation is very limited to the rainy season, as explained before. D spatiotemporal behaviour is consistent with the climatic regionalization of the study area, higher where accumulated rainfall reaches the highest values, in the Altiplano, and lower in elevated areas distant from the Amazon Basin, where the rainy season is not so evident.

These changes agree with the findings of Junquas et al. (2016) and Zappalà et al. (2018), regarding the position, strengthening and northward shift of the ITCZ (ascending branch of the Hadley cell) and its direct influence to intertropical rainfall patterns, so its influence will be less noticed in the intertropical area of Northern Chile. This could affect the coastal branch of the subtropical anticyclones and generate anomalies in the Walker circulation. These modifications on the atmospheric mechanisms alter the normal behaviour

of the South American monsoon described in Sarricolea and Romero (2015), and shifting northward the location of the Bolivian High in the high troposphere. The general warming trends identified in the region, more evident in the northern area (Meseguer-Ruiz et al., 2018b), can feed the convection activity, which, in result, would also intensify the Bolivian High. These two facts, northward shift and intensification, affect the Bolivian High and modify (intensify) the normal behaviour of the eastern fluxes that transport moisture

from the Amazon Basin (Segura et al., 2016), which would modify the temporal behaviour of precipitation as identified in this work, linking once again the temperature changes (warming) with modifications (intensification) of the hydrological cycle (Held and Soden, 2006). Thus, the dee point is reached in higher elevations, which can make that the previously referred convective processes that operates distinctly during day and night can be modified (Wasko et al., 2018). The regional-scale cyclonic circulation could be

strengthened during the afternoon and drive thermal circulations, agreeing with the results of Endries et al. (2018) and Junquas et al. (2018). This would not be happening in the west slope of the Andes and beneath 4 500 m a s l, where these processes have not been identified (Houston and Hartley, 2003; Junquas et al., 2013).

## 5. Conclusions

The spatial distribution of specific irregularity indices applied to precipitation temporal behaviour appears as a good tool to carry on this studies in arid and semiarid areas. While working with annual and monthly accumulated rainfall, the high degree of irregularity shown between years make climate projections have a very high degree of uncertainty. It is then more interesting to work considering different irregularity indices, so clear behaviours can be identified through time and space scales. These indices are well related with

geographical variables, such as the latitude, the elevation, the distance to the Amazon basin, the orientation and the curvature of the surface. This allows to interpolated and determine the spatial distribution of these indices based on these continuous spatial variables.

According to this, the C, representing the weight of the more rainy days within the whole rainy days, is high in the centre-south and lower in the north, which is representative of a very constricted rainy season.

C is increasing in the north, meaning that convective activity is getting more important in this area. The H index is showing more chaotic behaviour (noise) in the north with negative and significant trends and less in the south, but with positive and significant trends, which would be showing that the different patterns that originate precipitation in the study area could now act both in the north and south in opposite ways.





The $P_{11}$ index is high in the centre and south of the study area, where negative and significant trends appear, and low in the north with positive trends. Finally, the D index is showing positive and significant trends in the centre and south of the study area, where their values are the lower, so this means that precipitation conditions would be more related to frontal activity. The spatial and temporal behaviour of the studied indices are well related with the previously identified changes in temperature and atmospheric mechanisms, such as in the position of the ITCZ and the intensification of the eastern fluxes.

This spatiotemporal behaviour of precipitation agrees with evidenced changes in the last decades, and may be used to improve water management strategies and policies in arid and semiarid regions, where current GCM projections have a high uncertainty.

## 6. Author contributions

OMR conceived and coordinated the project, conducted the data analysis and trends and wrote the manuscript. JAG managed the database quality, homogenization and reconstruction. PPP performed the indices calculation. PS managed the multivariate regression models.

## 7. Competing interests

The authors declare that they have no conflict of interest.

## 8. Acknowledgements

The authors want to thank the FONDECYT Project 11160059, the CLICES Project (CGL2017-83866-C3-2-R) and the Climatology Group (2017 SGR 1362, Catalan Government) for their support.

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





**Table 1: Variables considered for the multivariate regression models.**

| Variable | Description | Abbreviation | Unit |
|---|---|---|---|
| Elevation | Elevation of each surface unit according to the Digital Elevation Model | $E_{lev}$ | Meters |
| Distance to the Amazon basin | Distance to the western limit of the Amazon basin | $D_{AB}$ | Degrees |
| Latitude | Distance from the 17ºS parallel | $L_{at}$ | Degrees |
| Orientation | Mean orientation of each surface unit | $O_{ri}$ | Degrees |
| Curvature | Mean curvature of each surface unit | $C_{urv}$ | Degrees |



**Table 2. Regression statistics for each regression model.**

| Index | Multiple correlation coefficient | Determination coefficient ($R^2$) | Adjusted $R^2$ | Typical error |
|---|---|---|---|---|
| C | 0,7218 | 0,5211 | 0,5048 | 0,0323 |
| H | 0,7837 | 0,6143 | 0,6011 | 0,6890 |
| $P_{11}$ | 0,6856 | 0,4701 | 0,4594 | 0,0216 |
| D | 0,7654 | 0,5858 | 0,5717 | 0,1152 |