# Peer review of "Spatial distribution and trends of different precipitation variability indices based on daily data in Northern Chile between 1966 and 2015"

_Hydrology and Earth System Sciences, 2018_

## Short Comment (SC1) · 13 Aug 2018

Captions in Figures 5 to 8: do you mean "confidence level"?

---

## Short Comment (SC2) · 13 Aug 2018

I understand your point, but I think it is not clear in the figure captions. Significance ($\alpha$) and confidence ($1 - \alpha$) are complementary. Did you use $\alpha = 10\%$? If that's the case then your significance level is $10\%$, and not $90\%$ as reported in the figure captions.

---

## Author Comment (AC1) · 13 Aug 2018

Thank you so much for you comment. Yes, we have decided to use the expression "Significance level" before than "Confidence level" to differentiate it from the more used "Confidence interval".

---

## Author Comment (AC2) · 13 Aug 2018

I do really get it. It has been a complete missunderstanding from our part. It will be modified. Again, thank you for your observation.

---

## Referee Comment (RC1) · Anonymous Referee #1 · 4 Sep 2018

The main objective of the manuscript is to measure four different precipitation indices, their tendencies over Northern Chile and their statistical link to geographical variables. In my opinion, there is a strong need in literature for articles assessing observed local precipitation climate change at specific networks of stations, since many current studies focus on coarser spatial scales derived from regional or global models, which are prone to large bias, particularly in case of rainfall. Thus, the aim of this work is certainly of interest for the readers of HESS.

The topics presented are covered with a good amount of material, and the network employed is of particular interest due to the high altitudes of many stations and the scarcity

of dataset covering arid regions. The manuscript is written in an understandable way, even if a revision from an native English speaker is recommended. Results are novel and interesting, exploiting a recently recovered large station network of particular geographical and climate relevance, and are well presented and explained. Therefore, I highly recommend the publication of the manuscript. Some steps of the analysis might not be easy to reproduce, given that three of the precipitation indices considered (Concentration Index, Entropy, Fractal dimension) are not straightforward to measure. However, even if their methodology could be further described, I think it is not in the scope of this paper, which already references the original articles describing these indices in detail.

In Section 2.1 ("Observed rainfall data"), authors describe the quality controls and homogenization employed. No station of the 161 employed was discarded, however, they specified that missing data were present for some stations. To better understand the degree of quality of the dataset, the authors could specify which was the maximum % of missing data measured in their stations during the study period.

In the Discussion Section, the authors compare their findings with those provided for other geographical areas already available in literature. In the case of the Concentration Index, the authors might also consider to include a comparison with the recently published work of Sangüesa et al. (2018)\* for Southern Chile. It is interesting to notice that the range of annual values of the Concentration Index in Northern Chile [0.42-0.67] is quite smaller than the range observed in Europe at annual scale [0.51-0.72] in the paper of Cortesi et al. (2012)\*\*, who employed exactly the same methodology to define the index, albeit with a slightly shorter period (1971-2010). At a first glance, such a range difference seems to be quite unlikely, due to the much more arid nature of Northern Chile compared to Europe, which should determine higher values of the Concentration Index. However, the lower values measured in Norther Chile might arise from a severely different gamma distribution of precipitation in desert climates; in fact, a lack of days with small or very small precipitation amount in the desert should

HESSD
determine an important change of the exponential curve of the Concentration Index, basically pushing the left part of the curve closer to the Equidistribution line, and explaining why in Norther Chile the values are lower than in Europe. I suggest the authors to introduce the comparison with Europe in the Discussion section, explaining why, in their opinion, the observed range in Northern Chile is smaller than the European one. To further improve the manuscript, the authors might want to include a similar comparison of the ranges of the values of the Entropy and Fractal dimension indices, particularly if important differences with the ranges measured in other continents are detected. Unfortunately I'm not proficient with these two indices, so I can't give any more suggestions to the authors.

The rest of the revision mainly addresses the readibility of the text, asks a few simple questions and provide grammatical corrections.

\* Sanguesa et al. (2018) Spatial and Temporal Analysis of Rainfall Concentration Using the Gini Index and PCI. Water 10(2), 112; https://doi.org/10.3390/w10020112

\*\* Cortesi at al. (2012) Daily precipitation concentration across Europe 1971–2010. Nat. Hazards Earth Syst. Sci., 12, 2799-2810, 2012; https://doi.org/10.5194/nhess-12-2799-2012

Page 1, line 11: replace sentence with this one: "Northern Chile is one of the most arid regions in the world, as it includes the Atacama Desert. At high latitudes, most of precipitation is observed only in a short period of the year, from December to March. This makes water availability one of the main concerns for policymakers".

Page 1, line 14: replace "and this makes that" with "for this reason,".

Page 1, line 18: replace the second "determined" in the sentence with a synonimous, e.g: "caused".

Page 1, line 24: replace the second "these" of the sentence with "this".

Page 2, line 1: the summer season mentioned is the Boreal or Austrual one?
Page 2, line 3: replace the sentence with "Such a configuration of the upper levels is known as Bolivian High (250 hPa). it activates the South Americano monsoon, (...). As a consecuence, the dry or wet characterisation (...).

Page 2, line 19: replace "prior" with "main".

Page 2, line 22: correct the sentence: (...) they show a particular degree (...). Such patterns increase at high latitudes and in wet regions (...).

Page 2, line 25: add an "s" after "work".

Page 2, line 28: remove "or even smaller".

Page 2, line 29: replace "This exposes that" with "Thus,".

Page 2, line 40: replace "study" with "studied".

Page 3, line 4: replace "face characterization of precipitation" with "characterize precipitation".

Page 3, line 6: replace "this" with "the".

Page 3, line 20: replace "this" with "these studies".

Page 4, lines 1-5: redundant, can be removed.

Page 5, line 21: replace "are" with "is" or remove "every".

Page 8, line 10: take advantage of this line to introduce in the text between parenthesis the symbols used at page 9 for referring to variables elevation, curvature, orientation and distance to the Amazon basin (they are still not defined in the paper).

Page 8, line 22: replace "significance" with "robustness".

Page 8, line 18: I can't find neither Table 1 neither Table 2 in the manuscript. Were they included?

Page 9, line 17: replace "(17) to (20)" with "(16) to (19)".
Page 12, line 10: replace "in" with "at".

Page 15, line 25: replace the first sentence with "In this work, the spatial distribution of specific irregularity indices applied to precipitation temporal behaviour has been presented, as it represent a good tool to carry on studies in arid and semiarid areas. In case of annual and monthly accumulated rainfall, the high degree of interannual variability determines a similar high degree of uncertainty of climate p projections. It is more interesting to consider (...) This allows to interpolate and measure (...).

Page 15, line 34: replace "very constricted" with "short".

Page 15, line 35: replace "C is increasing" with "C increases".

Page 16, line 1: expand the results: "The P11 index is mostly homogeneous in all the study region, with only a slight increase at the centre and south, (...)".

---

## Referee Comment (RC2) · Anonymous Referee #2 · 11 Sep 2018

This paper calculates a series of precipitation indices in Northern Chile and analyse their spatial distribution and trends over 1996-2015 period. There is some nice discussion on the different precipitation indices and their relationship with topography and climatic characteristics of the study region, which they relate with previous findings. However, the authors have not clearly elaborated how these analyses contribute to the objectives of the study (stated in the introduction). The manuscript cannot be read fluidly, and there are some logic sequences in the argument that are hard to follow. I think the topic can be of interest for HESS audience, however, the manuscript would require some major improvements. Below I provide some general comments, followed by a series of specific questions and recommendations.

General comments:

The authors generate a series of indices for each meteorological station, and then spatially distribute these indices based on interpolations techniques. Have the authors considered to use a gridded precipitation product instead, and then calculate the indices for each cell? I think is more physically robust since we have more information about the spatial distribution of precipitation phenomena (e.g., regional models, geostatistical interpolation). Can the authors elaborate on this?

There is a new platform with climate projections for the region that I think should be explored as part of the literature review of this work (http://simulaciones.cr2.cl).

Tables are placed at the end of the manuscript, while Figures are placed within the text. Please maintain consistency.

Tables: commas should not be used for decimals.

The complete manuscript should be revised since there are plenty of grammatical errors that do not permit a fluid reading.

Specific comments:

Abstract: "precipitation is recorded in a very constricted season every year". Is it that the recording is constricted or that the events are very rare? please clarify.

"Accumulated rainfall presents very high differences from one year to another, and this makes that climate projections have a very low degree of confidence in this area. So to this region it is more interesting to study the irregularity of precipitation itself instead of the accumulated rainfall values." The logic sequence of these sentences is not clear, please elaborate the point.

"These results will help to improve climate projections for these region and to inform the development of water management policies." It is not clear how climate projections and water management policies can use such results. Can the authors clarify? Also, Interactive comment

there is a typo in 'these region'.

P2L5-8: can the authors elaborate on the logic of these sentences?

P2L20: This sentence requires further discussion. There are studies aiming at differentiating these two effects (e.g., Boisier et al., 2016).

P2L21: there is literature available analysing projected changes in the region, which I think is more suitable than the cited work.

P2L21: should say 'even though'

P4L2-3: is not clear how the confidence of future projections may be more accurate by a better understanding on the temporal behaviour of precipitation. Can the author elaborate this point?

P4L3-5: the last sentence is not clear. Please re-phrase.

P5L2: "..." should not be used in scientific writing. Please re-phrase and correct typo ("do not varies")

P5L9: please provide references to support the statement: "an area where climate projections are not able to determine a clear trend for precipitation".

P5L11: is not clear how the determination of indices trends can be used to discriminate natural variability from anthropic forcing. Please elaborate and provide references.

P5L14-15: please correct grammatical errors.

Sect. 2.1: There is a missing citation for the meteorological stations (institution, where were the records obtained from, etc.). Also, general statistics on the record lengths (mean, min, max) could be provided, as well as the percentage of missing records. In addition, the elevation of meteorological stations should be provided (to have an idea of the representation of altiplanic zone).

Given the convective nature of precipitation events in this area, we expect to see many
"outliers", so I wonder if the quality control applied here is correct.

Can the authors explain how they fill missing data? they mention "undoing the normalisation of the reference series", but is not clear how the procedure is done, and under what assumptions. How much missing data is allowed?

The rest of sect. 2.1 is hard to understand, please improve the methodological description. Probably providing equations may be useful.

Sect. 2.3: There is a finer topographical resolution available (e.g., SRTM at 30-m), which I think is more suitable for this region (characterised by high elevation, but also low slopes in some sub-areas). Especially if this is the data used to calculate the gauge elevation.

Please correct the last sentence of the section.

Sect. 3.1: It is not clear how the regression coefficients are obtained. Please clarify.

Please correct the number of the equations.

Conclusions: Please correct grammatical errors.

P15L26: please elaborate the statement "the high degree of irregularity shown between years make climate projections have a very high degree of uncertainty:

P16L8: can the authors provide examples on how the generated information can be used to water management policies?

References: Boisier, J. P., R. Rondanelli, R. D. Garreaud, and F. Muñoz (2016), Anthropogenic and natural con- tributions to the Southeast Pacific precipitation decline and recent megadrought in central Chile, Geophys. Res. Lett., 43, 413–421, doi:10.1002/2015GL067265.

---

## Author Comment (AC3) · 14 Sep 2018

The main objective of the manuscript is to measure four different precipitation indices, their tendencies over Northern Chile and their statistical link to geographical variables. In my opinion, there is a strong need in literature for articles assessing observed local precipitation climate change at specific networks of stations, since many current studies focus on coarser spatial scales derived from regional or global models, which are prone to large bias, particularly in case of rainfall. Thus, the aim of this work is certainly of interest for the readers of HESS.

The authors would like to thank the reviewer for his work regarding the comments on this manuscript and his nice words.

The topics presented are covered with a good amount of material, and the network employed is of particular interest due to the high altitudes of many stations and the scarcity of dataset covering arid regions. The manuscript is written in an understandable way, even if a revision from a native English speaker is recommended. Results are novel and interesting, exploiting a recently recovered large station network of particular geographical and climate relevance, and are well presented and explained. Therefore, I highly recommend the publication of the manuscript. Some steps of the analysis might not be easy to reproduce, given that three of the precipitation indices considered (Concentration Index, Entropy, Fractal dimension) are not straightforward to measure. However, even if their methodology could be further described, I think it is not in the scope of this paper, which already references the original articles describing these indices in detail.

The authors appreciate the comments regarding the language. We hope that after the comments made by the author, it will be better understood. A more precise description of each index was not included for the same reason that the reviewer points.

In Section 2.1 ("Observed rainfall data"), authors describe the quality controls and homogenization employed. No station of the 161 employed was discarded; however, they specified that missing data were present for some stations. To better understand the degree of quality of the dataset, the authors could specify which was the maximum % of missing data measured in their stations during the study period.

Section 2.1 has been rewritten, defining the values of missing data and doing a better explanation of the filling method applied in this case.

In the Discussion Section, the authors compare their findings with those provided for other geographical areas already available in literature. In the case of the Concentration Index, the authors might also consider to include a comparison with the recently published work of Sangüesa et al. (2018)* for Southern Chile. It is interesting to notice that the range of annual values of the Concentration Index in Northern Chile [0.42-0.67] is quite smaller than the range observed in Europe at annual scale [0.51-0.72] in the paper of Cortesi et al. (2012)**, who employed exactly the same methodology to define the index, albeit with a slightly shorter period (1971-2010). At a first glance, such a range difference seems to be quite unlikely, due to the much more arid nature of Northern Chile compared to Europe, which should determine higher values of the Concentration Index. However, the lower values measured in Norther Chile might arise from a severely different gamma distribution of precipitation in desert climates; in fact, a lack of days with small or very small precipitation amount in the desert should determine an important change of the exponential curve of the Concentration Index, basically pushing the left part of the curve closer to the Equidistribution line, and explaining why in Norther Chile the values are lower than in Europe. I suggest the authors to introduce the comparison with Europe in the Discussion section, explaining why, in their opinion; the observed range in Northern Chile is smaller than the European one. To further improve the manuscript, the authors might want to include a similar comparison of the ranges of the values of the Entropy and Fractal dimension indices, particularly if important differences with the ranges measured in other continents are detected. Unfortunately, I'm not proficient with these two indices, so I can't give any more suggestions to the authors.

The reviewer points one key concept about these indices, and such differences showed here may have both climatological and geographical explanations, but also statistical ones. Some appointments have also been made for the other indices. The Discussion section has been developed as it follows:

"The range of annual values of the Concentration Index in Northern Chile (0.42-0.67) is quite smaller than the range observed in Europe at annual scale (0.51-0.72) in the work of Cortesi et al. (2012), who employed exactly the same methodology to define the index, albeit with a slightly shorter period (1971-2010). Such a range difference seems to be quite unlikely, due to the much more arid nature of Northern Chile compared to Europe, which should determine higher values of the Concentration Index. However, the lower values measured in Norther Chile might arise from a severely different gamma distribution of precipitation in desert climates. In fact, a lack of days with small or very small precipitation amount in the desert should

determine an important change of the exponential curve of the Concentration Index, basically pushing the left part of the curve closer to the equidistribution line, and explaining why in Northern Chile the values are lower than in Europe. The geographical and climatological reason of the reported differences could be the presence of the Mediterranean, which rises high sea surface temperatures during the autumn and triggers convective processes, leading to higher Concentration Index values."

The rest of the revision mainly addresses the readability of the text, asks a few simple questions and provide grammatical corrections.
The comments have been considered to improve the manuscript. We thank the reviewer for its effort.

\* Sanguesa et al. (2018) Spatial and Temporal Analysis of Rainfall Concentration Using the Gini Index and PCI. Water 10(2), 112; https://doi.org/10.3390/w10020112
\*\* Cortesi at al. (2012) Daily precipitation concentration across Europe 1971–2010. Nat. Hazards Earth Syst. Sci., 12, 2799-2810, 2012; https://doi.org/10.5194/nhess-12-2799-2012

Page 1, line 11: replace sentence with this one: "Northern Chile is one of the most arid regions in the world, as it includes the Atacama Desert. At high latitudes, most of precipitation is observed only in a short period of the year, from December to March. This makes water availability one of the main concerns for policymakers".
The text was corrected.

Page 1, line 14: replace "and this makes that" with "for this reason,".
The text was corrected.

Page 1, line 18: replace the second "determined" in the sentence with a synonimous, e.g: "caused".
The text was corrected.

Page 1, line 24: replace the second "these" of the sentence with "this".
The text was corrected.

Page 2, line 1: the summer season mentioned is the Boreal or Austrual one?
The text was corrected.

Page 2, line 3: replace the sentence with "Such a configuration of the upper levels is known as Bolivian High (250 hPa). it activates the South Americano monsoon, (...). As a consequence, the dry or wet characterisation (...).
The text was corrected.

Page 2, line 19: replace "prior" with "main".
The text was corrected.

Page 2, line 22: correct the sentence: (...) they show a particular degree (...). Such patterns increase at high latitudes and in wet regions (...).
The text was corrected.

Page 2, line 25: add an "s" after "work".
The text was corrected.

Page 2, line 28: remove "or even smaller".
The text was corrected.

Page 2, line 29: replace "This exposes that" with "Thus,".
The text was corrected.

Page 2, line 40: replace "study" with "studied".
The text was corrected.

Page 3, line 4: replace "face characterization of precipitation" with "characterize precipitation".
The text was corrected.

Page 3, line 6: replace "this" with "the".
The text was corrected.

Page 3, line 20: replace "this" with "these studies".
The text was corrected.

Page 4, lines 1-5: redundant, can be removed.
We believe that this sentence concludes the identification of the problems previously identified.

Page 5, line 21: replace "are" with "is" or remove "every".
The text was corrected.

Page 8, line 10: take advantage of this line to introduce in the text between parenthesis the symbols used at page 9 for referring to variables elevation, curvature, orientation and distance to the Amazon basin (they are still not defined in the paper).
The text was corrected. Despite this, the variables are characterized in Table 1 located in page 22.

Page 8, line 22: replace "significance" with "robustness".
The text was corrected.

Page 8, line 18: I can't find neither Table 1 neither Table 2 in the manuscript. Were they included?
Both tables appear after the reference list, in pages 22 and 23.

Page 9, line 17: replace "(17) to (20)" with "(16) to (19)".
The text was corrected.

Page 12, line 10: replace "in" with "at".
The text was corrected.

Page 15, line 25: replace the first sentence with "In this work, the spatial distribution of specific irregularity indices applied to precipitation temporal behaviour has been presented, as it represent a good tool to carry on studies in arid and semiarid areas. In case of annual and monthly accumulated rainfall, the high degree of interannual variability determines a similar high degree of uncertainty of climate p projections. It is more interesting to consider (...) This allows to interpolate and measure (...).
The text was corrected.

Page 15, line 34: replace "very constricted" with "short".
The text was corrected.

Page 15, line 35: replace "C is increasing" with "C increases".
The text was corrected.

Page 16, line 1: expand the results: "The P11 index is mostly homogeneous in all the study region, with only a slight increase at the centre and south, (...)".
The text was corrected.

---

## Referee Comment (RC3) · Anonymous Referee #1 · 20 Sep 2018

To be able to reply to the Authors, I need to take a look at the revised version of their Manuscript. Is it already available online?

---

## Referee Comment (RC4) · Anonymous Referee #1 · 21 Sep 2018

The Authors addressed all the main issues of my review. A few final remarks:

- given the high % of missing data, you might flag stations with more than 50% of missing data with a different color or symbol in figure 1. - remove the decimals beyond the second in the legend figures 5-8. - did you compare your results with those of Sanguesa et al. (2018)? - add the reference to the paper of Cortesi et al. (2012) in the bibliography

Grammar revision:

- Page 1 line 15: replace the first "to" with "in" - Page 1 line 19: replace "caused" with "measured" - Page 1 line 19: remove 'determined' and add 'was determined' after

'distribution' - Page 1 line 25 and 26: replace 'doing' with a more appropriate verb - Page 2 line 8: add 'in' after 'recorded' - Page 2 line 24: remove 'but' - Page 3 line 9: I don't understand very well the new sentence - Page 10 line 2: replace 'by mean' with 'by means'

---

## Author Comment (AC4) · 21 Sep 2018

This paper calculates a series of precipitation indices in Northern Chile and analyse their spatial distribution and trends over 1996-2015 period. There is some nice discussion on the different precipitation indices and their relationship with topography and climatic characteristics of the study region, which they relate with previous findings. However, the authors have not clearly elaborated how these analyses contribute to the objectives of the study (stated in the introduction). The manuscript cannot be read fluidly, and there are some logic sequences in the argument that are hard to follow. I think the topic can be of interest for HESS audience, however, the manuscript would require some major improvements. Below I provide some general comments, followed by a series of specific questions and recommendations.

The authors would like to thank the reviewer for his work regarding the comments on this manuscript and his nice words. The authors appreciate the comments regarding the language. We hope that after the comments made by the reviewers, it will be better understood.

**General comments:**

The authors generate a series of indices for each meteorological station, and then spatially distribute these indices based on interpolations techniques. Have the authors considered to use a gridded precipitation product instead, and then calculate the indices for each cell? I think is more physically robust since we have more information about the spatial distribution of precipitation phenomena (e.g., regional models, geostatistical interpolation). Can the authors elaborate on this?

The reviewer rises one key topic worked here. In this case, we decided to work only with observed data as a part of the goal of the manuscript and then just see how geographical factors influence on it. We totally agree on the idea of the interest to work with interpolated precipitation data, but for daily data resolution, we bet to work only with meteorological stations in this case.

There is a new platform with climate projections for the region that I think should be explored as part of the literature review of this work (http://simulaciones.cr2.cl).

We totally agree that there is a great work done by the CR2 team according to climate change projections. It has been mentioned in the introduction.

**Tables are placed at the end of the manuscript, while Figures are placed within the text. Please maintain consistency.**

The guidelines for the manuscript preparation of HESS says that Tables "should appear on separate sheets after the references and should be numbered sequentially with Arabic numerals", but it's not the case for the Figures, that's why we placed them as they appear.

**Tables: commas should not be used for decimals.**

Table 2 has been modified accordingly.

The complete manuscript should be revised since there are plenty of grammatical errors that do not permit a fluid reading.

The grammatical mistakes have been corrected, we hope that now it will be easy to read.

**Specific comments:**

Abstract: "precipitation is recorded in a very constricted season every year". Is it that the recording is constricted or that the events are very rare? Please clarify.

We meant that the season was very short. It has been clarified in the text.

"Accumulated rainfall presents very high differences from one year to another, and this makes that climate projections have a very low degree of confidence in this area. So to this region it is more interesting to study the irregularity of precipitation itself instead of the accumulated rainfall values." The logic sequence of these sentences is not clear, please elaborate the point. It has been modified in order to make it clearer.

"These results will help to improve climate projections for these region and to inform the development of water management policies." It is not clear how climate projections and water management policies can use such results. Can the authors clarify? Also, there is a typo in 'these region'.

The last sentence has been expanded in order to do it more understandable. The typo has been corrected.

P2L5-8: can the authors elaborate on the logic of these sentences?

The text has been corrected to clarify the meaning of these sentences.

P2L20: This sentence requires further discussion. There are studies aiming at differentiating these two effects (e.g., Boisier et al., 2016).

The authors agree that the original sentence was not correct, while we were referring to precipitation projections in tropical regions. It has been clarified.

P2L21: there is literature available analysing projected changes in the region, which I think is more suitable than the cited work.

New references have been added:

- Baez-Villanueva, O. M., Zambrano-Bigiarini, M., Ribbe, L., Nauditt, A., Giraldo-Osorio, J. D., and Thinh, N. X.: Temporal and spatial evaluation of satellite rainfall estimates over different regions in Latin-America, Atmos. Res., 213, 34-50, https://doi.org/10.1016/j.atmosres.2018.05.011, 2018.

- Cabré, M. F., Solman, S., and Núñez M.: Regional climate change scenarios over southern South America for future climate (2080-2099) using the MM5 Model. Mean, interannual variability and uncertainties, Atmosfera, 29(1). 35-60, https://doi.org/10.20937/ATM.2016.29.01.04, 2016.

- Kitoh, A., Kusunoki, S., and Nakaegawa, T.: Climate change projections over South America in the late 21st century with the 20 and 60 km mesh Meteorological Research Institute atmospheric general circulation model (MRI-AGCM), J. Geophys. Res., 116, D06105, https://doi.org/10.1029/2010JD014920, 2011.

- Williams, C. J. R.: Climate Change in Chile: An Analysis of State-of-the-Art Observations, Satellite-Derived Estimates and Climate Model Simulations, J. Earth Sci. Clim. Change, 8, 5, https://doi.org/10.4172/2157-7617.1000400, 2017.

P2L21: should say 'even though'

It has been corrected.

P4L2-3: is not clear how the confidence of future projections may be more accurate by a better understanding on the temporal behaviour of precipitation. Can the author elaborate this point? It has been clarified.

P4L3-5: the last sentence is not clear. Please re-phrase. It has been rephrased.

P5L2: "..." should not be used in scientific writing. Please re-phrase and correct typo ("do not varies") It has been corrected.

P5L9: please provide references to support the statement: "an area where climate projections are not able to determine a clear trend for precipitation". We added the reference "IPCC, 2013".

P5L11: is not clear how the determination of indices trends can be used to discriminate natural variability from anthropic forcing. Please elaborate and provide references.

While it is not the goal of the work developed here, we have rephrased the sentence. We added "This can be determined by comparing the behaviour of these precipitation variability indices with the behaviour of temperatures, linking their influence on the hydrological cycle."

P5L14-15: please correct grammatical errors. They have been corrected.

Sect. 2.1: There is a missing citation for the meteorological stations (institution, where were the records obtained from, etc.). Also, general statistics on the record lengths (mean, min, max) could be provided, as well as the percentage of missing records. In addition, the elevation of meteorological stations should be provided (to have an idea of the representation of altiplanic zone).

The institutions that provide the data and the link from they were obtained have been added: "The data are obtained from the database provided by the by the Chilean Water Directorate (DGA) and Chilean Meteorological Directorate (DMC) and available online (http:// http://www.cr2.cl/bases-de-datos/)."

The mean annual amounts can be seen in Fig. 2, and some further detail of the meteorological stations has been added as an Annex.

Given the convective nature of precipitation events in this area, we expect to see many "outliers", so I wonder if the quality control applied here is correct.

Can the authors explain how they fill missing data? They mention "undoing the normalisation of the reference series", but is not clear how the procedure is done, and under what assumptions. How much missing data is allowed?

Section 2.1 has been rewritten, defining the values of missing data and doing a better explanation of the filling method applied in this case:

Daily rainfall records from 161 stations across the study area for the period 1st January 1966 to 31st December 2015 are used, with a varied number of missing data (a quarter of the series had less than 19.8% of missing data, half of them had less than 46%, and the other quarter had more than 73.5%). The data are obtained from the database provided by the by the Chilean Water Directorate (DGA) and Chilean Meteorological Directorate (DMC) and available online (http://http://www.cr2.cl/bases-de-datos/). Further information of the meteorological stations can be found in the supplementary material (Annex A).

The quality control was developed using the R package Climatol version 3.0 (Guijarro, 2016), which uses normal ratio values (every data is divided by the mean of its series) of the closest precipitation data to build reference series for all the stations. In this way, every precipitation data  $P_{ik}$  of series *i* is "normalized" by  $\widehat{p_{ik}} = P_{ik}/\widehat{P}_{i}$ , and then all data (whether existing or missing) is estimated as:

$$\widehat{p_{ik}} = \frac{\sum_{j=1}^{j=n} w_{ij} p_{jk}}{\sum_{j=1}^{j=n} w_{ij}}$$
(1)

in which  $\widehat{p_{lk}}$  is the estimated precipitation from the nearest *n* data in time step *k*, weighted by  $w_{ij}$ . Averages  $\overline{P_l}$  are computed initially with the available data, but new averages are obtained after filling all missing data with their estimates, repeating the process until no average differs more than 0.05 mm from its value at the previous iteration. Differences between observed and reference series (in normalized form) are then used to test their quality by outlier detection, and also to check their homogeneity through the SNHT test (Alexandersson, 1986), using up to n = 10 reference data in order to smooth any possible inhomogeneities in the nearby stations. The detection of significant shifts in the mean was done on the monthly aggregates of the series, since the much higher variability of the daily series makes that detection far more difficult, especially in such arid climates as in the studied area.

The rest of sect. 2.1 is hard to understand, please improve the methodological description. Probably providing equations may be useful.

Please, check previous response.

**Sect. 2.3: There is a finer topographical resolution available (e.g., SRTM at 30-m), which I think is more suitable for this region (characterised by high elevation, but also low slopes in some sub-areas). Especially if this is the data used to calculate the gauge elevation.**

The reviewer is totally right, but the authors decided to work with a wider resolution because the results won't change significantly and the computing times are consequently lower. Moreover, the elevation considered for the calculation were the elevations provided by DGA and DMC.

Please correct the last sentence of the section. The sentence has been corrected.

Sect. 3.1: It is not clear how the regression coefficients are obtained. Please clarify.

We have added, in section 3.1, the way the regression coefficients are obtained:

"The multivariate regression models are linear, and the coefficients are obtained by mean of the minimum mean square error method".

Please correct the number of the equations. The text has been corrected.

Conclusions: Please correct grammatical errors. The text has been corrected.

P15L26: please elaborate the statement "the high degree of irregularity shown between years make climate projections have a very high degree of uncertainty: The sentence has been rephrased.

P16L8: can the authors provide examples on how the generated information can be used to water management policies?

We added the sentence: "Depending on the water availability, the development of economic activities can be adjusted so water supply may be guaranteed for the whole community."

References: Boisier, J. P., R. Rondanelli, R. D. Garreaud, and F. Muñoz (2016), Anthropogenic and natural contributions to the Southeast Pacific precipitation decline and recent megadrought in central Chile, Geophys. Res. Lett., 43, 413–421, doi: 10.1002/2015GL067265.

---

## Author Comment (AC5) · 21 Sep 2018

The comment was uploaded in the form of a supplement:
https://www.hydrol-earth-syst-sci-discuss.net/hess-2018-371/hess-2018-371-AC5-supplement.pdf

---

## Author Comment (AC6) · 21 Sep 2018

[revised manuscript text omitted]

**2. Material and methods**

**2.1. Observed rainfall data**

Daily rainfall records from 161 stations across the study area for the period 1st January 1966 to 31st December 2015 are used, with a varied number of missing data (a quarter of the series had less than 19.8% of missing data, half of them had less than 46%, and the other quarter had more than 73.5%). The data are obtained from the database provided by the by the Chilean Water Directorate (DGA) and Chilean Meteorological Directorate (DMC) and available online (http:// http://www.cr2.cl/bases-de-datos/). Further information of the meteorological stations can be found in the supplementary material (Annex A).

The quality control was developed using the R package Climatol version 3.0 (Guijarro, 2016), which uses normal ratio values (every data is divided by the mean of its series) of the closest precipitation data to build reference series for all the stations. In this way, every precipitation data $P_{ik}$ of series $i$ is "normalized" by $\widehat{p_{ik}} = P_{ik}/\widehat{P_i}$, and then all data (whether existing or missing) is estimated as:

$$\widehat{p_{ik}} = \frac{\sum_{j=1}^{j=n} w_{ij} p_{jk}}{\sum_{j=1}^{j=n} w_{ij}} \tag{1}$$

in which $\widehat{p_{ik}}$ is the estimated precipitation from the nearest $n$ data in time step $k$, weighted by $w_{ij}$. Averages $\overline{P_i}$ are computed initially with the available data, but new averages are obtained after filling all missing data with their estimates, repeating the process until no average differs more than 0.05 mm from its value at the previous iteration. Differences between observed and reference series (in normalized form) are then used to test their quality by outlier detection, and also to check their homogeneity through the SNHT test (Alexandersson, 1986), using up to $n = 10$ reference data in order to smooth any possible inhomogeneities in the nearby stations. 
[revised manuscript text omitted]
 ($E_{lev}$), curvature ($C_{urv}$) and orientation ($O_{ri}$) (Ninyerola et al., 2000; 2007), the latitude ($L_{at}$) and the distance to the Amazon basin ($D_{AB}$). The spatial resolution of these continuous variables has a spatial resolution of 90 m x 90 m.

Attending that multivariate regression models generate residuals spatially distributed (differences between the meteorological stations and the values determined by the models), they were corrected following an Inverse Distance Weighting interpolation (IDW) with a power magnitude of 2 (Chen et al., 2017). This allows to obtain interpolations that approach in a better way to the spatial heterogeneity, by decreasing the mistake of the model and deleting the outliers (Crespi et al., 2018).

The multivariate regression models will be expressed according to the variables shown in Table 1, and a constant.

**2.4. Variability indices trends**

The analysis was conducted for the period 1966-2015; therefore, the annual variability indices values found were obtained over a broad period of 50 years, which gives climatic robustness to the results. The trends are determined applying the Mann-Kendall non-parametric test (Mann, 1945; Kendall, 1962). The MK statistic is calculated as:

$$S = \sum_{i=1}^{n-1} \sum_{j=i+1}^{n} sgn(x_j - x_i) \tag{12}$$

$$(x_j - x_i) = z \tag{13}$$

$$sgn(z) = \begin{cases} 1 \text{ if } (z) \geq 0 \\ 0 \text{ if } (z) = 0 \\ -1 \text{ if } (z) \leq 0 \end{cases} \tag{14}$$

Where $n$ is the dimension of the series and $x_j$ and $x_i$ are the annual values, respectively, in the years $j$ and $i$, with $j > i$. For $n > 10$, given that $x_i$ is an independent and randomly ordered series, the statistic $S$ follows a normal distribution whose mean is equal to 0, and variance given by:

$$Var(S) = [n(n-1)(2n+5) \sum_{i=1}^{n} t_i i(i-1)(2i+5)]/18 \tag{15}$$

Where $t_i$ represents a margin of error of $i$.

The standardized statistical test $Z_{MK}$ follows a standard normal distribution, and is represented by:

$$Z_{MK} = \begin{cases} \frac{S-1}{\sqrt{Var(S)}} \text{ if } S > 0 \\ 0 \text{ if } S = 0 \\ \frac{S+1}{\sqrt{Var(S)}} \text{ if } S < 0 \end{cases} \tag{16}$$

Using a two-tailed test, if $Z_{MK}$ is greater than $Z_{(\alpha/2)}$, with a significance level $\alpha$, then it is possible to reject the null hypothesis and the trend can be considered significant.

**3. Results**

**3.1. Multivariate regression models**

The multivariate regression models are linear, and the coefficients are obtained by mean of the minimum mean square error method. They 
[revised manuscript text omitted]
., 2015). The range of annual values of the Concentration Index in Northern Chile (0.42-0.67) is quite smaller than the range observed in Europe at annual scale (0.51-0.72) in the work of Cortesi et al. (2012), who employed exactly the same methodology to define the index, albeit with a slightly shorter period (1971-

2010). Such a range difference seems to be quite unlikely, due to the much more arid nature of Northern Chile compared to Europe, which should determine higher values of the Concentration Index. However, the lower values measured in Northern Chile might arise from a severely different gamma distribution of precipitation in desert climates. In fact, a lack of days with small or very small precipitation amount in the desert should determine an important change of the exponential curve of the Concentration Index, basically pushing the left part of the curve closer to the equidistribution line, and explaining why in Northern Chile the values are lower than in Europe. The geographical and climatological reason of the reported differences could be the presence of the Mediterranean, which rises high sea surface temperatures during the autumn and triggers convective processes, leading to higher Concentration Index values. 
[revised manuscript text omitted]

---

## Author Comment (AC7) · 8 Oct 2018

The Authors addressed all the main issues of my review.

The authors would like to thank the reviewer for his work regarding the comments on this manuscript and his nice words. We agree with all the of them and we think that those will help to give a more accurate work. We also hope that the reviewer will now be satisfied with the last version.

A few final remarks:
- given the high % of missing data, you might flag stations with more than 50% of missing data with a different color or symbol in figure 1.

It has been done as proposed. Stations with more than 50% of missing data are marked with a cross instead of with a dot.

- remove the decimals beyond the second in the legend figures 5-8.

It has been done as proposed. The new figures have been added to the manuscript.

- did you compare your results with those of Sanguesa et al. (2018)?

Sangüesa et al. (2012) have proposed a very interesting work applied to Central Chile applying the Gini index to daily precipitation. However, to these author's minds, the work lacks of a good methodological approach. It has been discussed in another recently published work (Sarricolea et al., 2019). Moreover, the methodology and the study area are different from those considered in the present manuscript.

- add the reference to the paper of Cortesi et al. (2012) in the bibliography

It has been done as proposed.

Grammar revision:
- Page 1 line 15: replace the first "to" with "in"

It has been corrected.

- Page 1 line 19: replace "caused" with "measured"

It has been modified.

- Page 1 line 19: remove 'determined' and add 'was determined' after 'distribution'

It has been done as proposed.

- Page 1 line 25 and 26: replace 'doing' with a more appropriate verb

It has been replaced by "helping to develop".

- Page 2 line 8: add 'in' after 'recorded'

It has been added.

- Page 2 line 24: remove 'but'

It has been removed.

- Page 3 line 9: I don't understand very well the new sentence

The authors tried to emphasize the importance of considering such many aspects of both spatial and temporal behaviour of precipitation, not only considering monthly/daily accumulated.

- Page 10 line 2: replace 'by mean' with 'by means'

It has been done, in line 13, page 10.

---

## Author Comment (AC8) · 8 Oct 2018

The comment was uploaded in the form of a supplement:
https://www.hydrol-earth-syst-sci-discuss.net/hess-2018-371/hess-2018-371-AC8-supplement.pdf

---

## Author Comment (AC9) · 8 Oct 2018

The comment was uploaded in the form of a supplement:
https://www.hydrol-earth-syst-sci-discuss.net/hess-2018-371/hess-2018-371-AC9-supplement.pdf